# Early triage of critically ill COVID-19 patients using deep learning

Wenhua Liang [1,12], Jianhua Yao[2,12], Ailan Chen[1,3,12], Qingquan Lv[3,12], Mark Zanin[4,12], Jun Liu [1,5,12], SookSan Wong[1], Yimin Li [6], Jiatao Lu[3], Hengrui Liang [1,5], Guoqiang Chen[7], Haiyan Guo[7], Jun Guo[8], Rong Zhou[1], Limin Ou[1], Niyun Zhou[2], Hanbo Chen[2], Fan Yang[2], Xiao Han[2], Wenjing Huan[9], Weimin Tang[9], Weijie Guan[1], Zisheng Chen[1,10], Yi Zhao[1], Ling Sang[1], Yuanda Xu[6], Wei Wang[5], Shiyue Li[1], Ligong Lu[11], Nuofu Zhang[1], Nanshan Zhong[1✉], Junzhou Huang[2✉] & Jianxing He [1✉]

The sudden deterioration of patients with novel coronavirus disease 2019 (COVID-19) into critical illness is of major concern. It is imperative to identify these patients early. We show that a deep learning-based survival model can predict the risk of COVID-19 patients developing critical illness based on clinical characteristics at admission. We develop this model using a cohort of 1590 patients from 575 medical centers, with internal validation performance of concordance index 0.894 We further validate the model on three separate cohorts from Wuhan, Hubei and Guangdong provinces consisting of 1393 patients with concordance indexes of 0.890, 0.852 and 0.967 respectively. This model is used to create an online calculation tool designed for patient triage at admission to identify patients at risk of severe illness, ensuring that patients at greatest risk of severe illness receive appropriate care as early as possible and allow for effective allocation of health resources.

[1] China State Key Laboratory of Respiratory Disease and National Clinical Research Center for Respiratory Disease, The First Affiliated Hospital of Guangzhou Medical University, Guangzhou, China. [2] Tencent AI Lab, Shenzhen, China. [3] Hankou Hospital, Wuhan, China. [4] School of Public Health, The University of Hong Kong, Hong Kong SAR, China. [5] Department of Thoracic Surgery, The First Affiliated Hospital of Guangzhou Medical University, Guangzhou, China. [6] Department of Intensive Care Unit, The First Affiliated Hospital of Guangzhou Medical University, Guangzhou, China. [7] Foshan Hospital, Foshan, China. [8] Daye Hospital, Hubei, China. [9] Tencent Healthcare, Shenzhen, China. [10] Department of Respiratory Disease, The Sixth Affiliated Hospital of Guangzhou Medical University, Guangzhou, China. [11] Zhuhai People Hospital, Zhuhai, China. [12] These authors contributed equally: Wenhua Liang, Jianhua Yao, Ailan Chen, Qingquan Lv, Mark Zanin, Jun Liu. ✉email: nanshan@vip.163.com; joehhuang@tencent.com; drjianxing.he@gmail.com

With coronavirus disease 2019 (COVID-19) now a pandemic, rapid and effective triage is critical for early treatment and effective allocation of hospital resources. COVID-19 disease has shown the worrying trend of sudden progression to critical illness in 6.5% of cases and with a mortality rate of 49% in these patients[1,2]. The influx of additional health resources in Hubei province, which was the epicenter of the outbreak, greatly improved patient outcomes. Since early intervention is associated with improved prognosis, the ability to identify patients that are most at risk of developing severe disease upon admission will ensure that these patients receive appropriate care as soon as possible.

Clinical researchers have been using survival analysis (also called time-to-event analysis) to estimate the probability of prognostic clinical outcomes such as death and cancer recurrence in the course of disease development and to plan optimal treatment schemes accordingly. The Cox proportional hazards model (CPH)[3] is a widely used statistical model that relies on regression analysis to determine the association between a predictor covariate, such as clinical characteristics, with the risk of an event occurring (e.g. "death"). The model assumes that the risk of an event is a linear combination of the patient's covariates, which may be too simplistic for some complex clinical events such as progression to critical illness.

The increase in computing power and the availability of big data has enabled deep learning to be used successfully in many medical applications[4]. For instance, convolutional neural networks, a form of deep learning, could detect skin cancers as effectively as dermatologists[5]. Deep learning could also successfully interpret pathology results to diagnose prostate cancer and basal cell carcinoma[6]. Deep neural networks have also been used to recommend personal treatment plans[7]. In this study, we integrate deep learning techniques with the traditional Cox model for survival analysis of the nonlinear effect from clinical covariates to predict clinical outcome of COVID-19 patients. We demonstrate that this Deep Learning Survival Cox model can efficiently triage COVID-19 patients with high accuracy.

## Results

**Data sources and characteristics.** On behalf of the National Clinical Research Center for Respiratory Disease and in collaboration with the National Health Commission (NHC) of the People's Republic of China, we established a retrospective cohort to study COVID-19 cases throughout China. We obtained medical records and compiled the data from laboratory-confirmed hospitalized cases with COVID-19 reported to the NHC between 21 November 2019 and 31 January 2020. The NHC requested that all of the 1855 designated hospitals for COVID-19 submit clinical records to the database. Hospitals whose clinical records had not been submitted by this deadline were requested again. Our cohort largely represents the overall situation as of 31 January, taking into account the proportion of hospitals (~one-third) and patient number (17.2%, 1590/9252 cases), as well as the broad coverage (covering 31 of 34 provinces/autonomous regions (appendix illustrated the geographic distribution of cases from all hospitals that contributed to the database)), although the non-responsive bias cannot be fully excluded.

Confirmed cases of COVID-19 were defined as patients who tested positive by high-throughput sequencing or real-time reverse-transcription PCR assay on nasal and pharyngeal swab specimens. Only laboratory-confirmed cases were included in our analysis. Critical illness was defined as a composite event of admission to an intensive care unit or requiring invasive ventilation, or death.

Our model training cohort included 1590 patients, of which 131 developed critical illness, from 575 medical centers (Supplementary Tables 1 and 2, and Appendix). To test the generalization of our model, we collected three independent cohorts as external validation sets with wide geographic coverage, one from a hospital in the epicenter Wuhan (940 patients, 94 critically ill), one from multiple centers in ten cities in Hubei province, excluding Wuhan (380 patients, 9 critically ill), and another from a hospital in Guangdong province, representing a province not suffering from the health resource burnout experienced in Wuhan (73 patients, 3 critically ill) (Supplementary Tables 3–5).

**Selection of critical illness predictors and model establishment.** Seventy-four baseline clinical features with at least 60% data completeness were considered as critical illness predictors and were used for model establishment. Ten features with statistically significant ($P < 0.05$) hazard ratios were identified through a machine learning variable selection algorithm called least absolute shrinkage and selection operator (LASSO)[8]. These were X-ray abnormalities, age, dyspnea, COPD (chronic obstructive pulmonary disease), number of comorbidities, cancer history, neutrophil/lymphocytes ratio, lactate dehydrogenase, direct bilirubin, and creatine kinase (Table 1).

**Table 1 Univariate analysis of the selected features for COVID-19 patients in the training cohort.**

**Critical illness**

| | Total ($n = 1590$) | No ($n = 1459$) | Yes ($n = 131$) | Hazard ratio (95% CI) | p-value | AUC (95% CI) | C-index (95% CI) |
|---|---|---|---|---|---|---|---|
| Age | 48.9 ± 15.7 | 47.8 ± 15.2 | 61.6 ± 14.8 | 1.059 (1.046–1.071) | <0.001 | 0.755 (0.695–0.812) | 0.732 (0.674–0.790) |
| Dyspnea | 331/1394 (23.7) | 257/1275 (20.2) | 74/119 (62.2) | 5.759 (3.973–8.346) | <0.001 | 0.665 (0.590–0.745) | 0.659 (0.584–0.739) |
| Cancer history | 18/1590 (1.1) | 11/1459 (0.8) | 7/131 (5.3) | 5.927 (2.766–12.7) | <0.001 | 0.498 (0.495–0.500) | 0.498 (0.495–0.500) |
| COPD | 24/1590 (1.5) | 12/1459 (0.8) | 12/131 (9.2) | 7.471 (4.124–13.53) | <0.001 | 0.532 (0.495–0.580) | 0.516 (0.495–0.549) |
| No. of comorbidity | | | | 1.67 (1.506–1.851) | <0.001 | 0.697 (0.613–0.789) | 0.682 (0.597–0.772) |
| 0 | 1191/1590 (74.9) | 1137/1459 (77.9) | 54/131 (41.2) | | | | |
| 1 | 269/1590 (16.9) | 229/1459 (15.7) | 40/131 (30.5) | | | | |
| 2 | 88/1590 (5.5) | 68/1459 (4.7) | 20/131 (15.3) | | | | |
| 3 | 34/1590 (2.1) | 20/1459 (1.4) | 14/131 (10.7) | | | | |
| 4 | 5/1590 (0.3) | 4/1459 (0.3) | 1/131 (0.8) | | | | |
| 5 | 3/1590 (0.2) | 1/1459 (0.1) | 2/131 (1.5) | | | | |
| X-ray abnormality | 243/1590 (15.3) | 184/1459 (12.6) | 59/131 (45) | 5.315 (3.765–7.504) | <0.001 | 0.600 (0.524–0.681) | 0.614 (0.535–0.696) |
| Neutrophil/lymphocytes | 5.1 ± 5.6 | 4.4 ± 3.8 | 12.7 ± 12.4 | 1.061 (1.052–1.071) | <0.001 | 0.861 (0.803–0.918) | 0.857 (0.795–0.914) |
| Lactate dehydrogenase | 314.3 ± 693.7 | 273.6 ± 135.2 | 723.6 ± 2239.5 | 1 (1–1) | <0.001 | 0.810 (0.745–0.868) | 0.787 (0.727–0.846) |
| Direct bilirubin | 4 ± 2.7 | 3.7 ± 2.3 | 6.5 ± 4.1 | 1.212 (1.165–1.261) | <0.001 | 0.674 (0.567–0.782) | 0.662 (0.551–0.773) |
| Creatine kinase | 135.5 ± 246.7 | 123 ± 125.3 | 258.9 ± 702.8 | 1.001 (1–1.001) | <0.001 | 0.557 (0.449–0.665) | 0.554 (0.441–0.666) |

Data are mean ± SD, n/N (%), where N is the total number of patients with available data. P-values are calculated by log rank test.

**Performance of the prediction model**. We divided the training cohort into 80% for model training and 20% for internal model validation with balanced data distribution. The concordance index (C-index, a standard performance metric for survival analysis) and area under the receiver-operator characteristic curve (AUC, a performance measurement for classification problem) were evaluated on the model validation cohort to assess discriminative ability. The C-index and AUC of our Deep Learning Survival Cox model were 0.894 (0.95 confidence interval (CI), 0.857–0.930) and 0.911 (0.95 CI, 0.875–0.945), respectively, on the model validation set, whereas those of the classic Cox model were 0.876 (0.95 CI, 0.830–0.921) and 0.889 (0.95 CI, 0.843–0.934), respectively (Fig. 1a). The predictive value of this model was higher than the CURB-6 model[9], with a C-index of 0.75 (95% CI, 0.70–0.80). The precision-Recall curves for the internal validation set is shown in Supplementary Fig. 1.

**Risk stratification**. We further calculated the risk of each individual in the entire training cohort and divided all patients into three groups based on the risk cut-off at 95% sensitivity and 95% specificity. A total of 875, 560, and 155 patients were classified in low-, medium-, and high-risk group, respectively, with the actual risk probability of critical illness events at 0.9%, 7.3%, and 52.9%, respectively. Kaplan–Meier curves of these three patient groups demonstrated statistically significant separation (Fig. 1b).

**External validation**. To test the generalization of this model, we tested the model performance on three independent cohort from different locations and with different health resource levels. The first cohort was from the epicenter Wuhan, the second from an area outside of Wuhan in Hubei province, and the last was from Guangdong province, a province that was not suffering from health resource burnout. The Wuhan cohort consisted of COVID-19 patients admitted in January and February (without overlap with the training set) to Hankou hospital, the Hubei cohort consisted of cases from multiple centers in ten cities before 31 January (which did not overlap with the training set), and the Guangdong cohort that included cases admitted between January and February to Foshan hospital. Data-processing procedures were identical to those used for the training cohort. Table 2 and

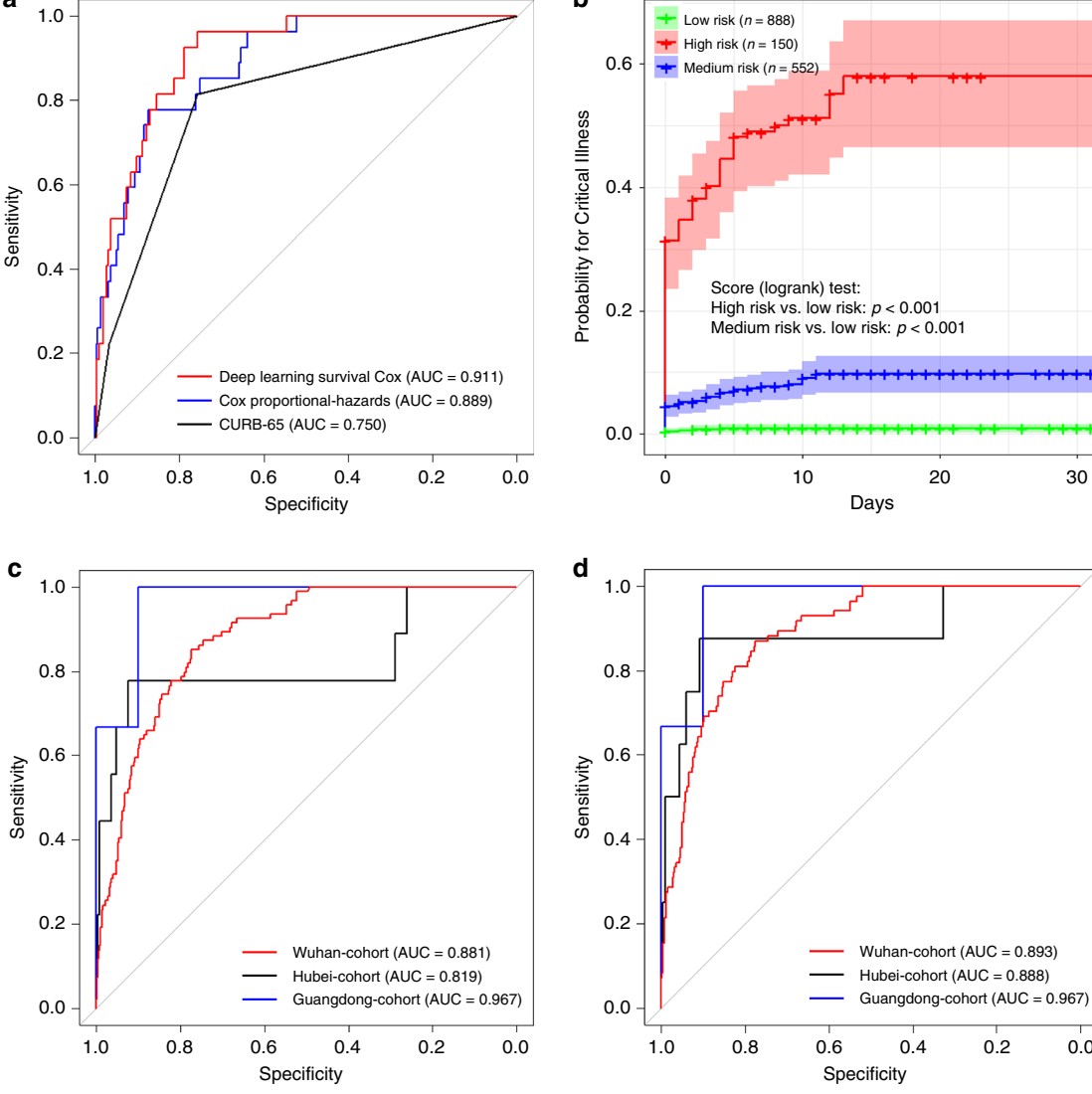

**Fig. 1 Model performance comparison. a** Comparison of ROC curves for the Deep Learning Survival Cox model and the Cox proportional hazards model on the training-validation set. **b** The Kaplan–Meier curves for developing critical illness among patients in different risk groups in the training set. Shaded areas indicate 95% confidence interval. **c** ROC curves for the three external validation cohorts using the entire datasets. **d** ROC curves for the three independent external validation cohorts, excluding patients that were missing more than three values.

**Table 2 Results of Deep Learning Survival Cox analyses on the three independent external validation cohorts.**

| Cohort | Wuhan | | Hubei | | Guangdong | |
|---|---|---|---|---|---|---|
| | Ex3 | All cases | Ex3 | All cases | Ex3 | All cases |
| No. of patients (critically) | 801 (84) | 940 (94) | 305 (8) | 380 (9) | 73 (3) | 73 (3) |
| AUC (95% CI) | 0.893 (0.867–0.919) | 0.881 (0.854–0.905) | 0.888 (0.732–0.984) | 0.819 (0.632–0.978) | 0.967 (0.905–1.000) | 0.967 (0.905–1.000) |
| C-index (95% CI) | 0.890 (0.865–0.915) | 0.878 (0.852–0.903) | 0.852 (0.672–0.973) | 0.769 (0.556–0.966) | 0.967 (0.906–1.000) | 0.967 (0.906–1.000) |
| HR&MR recall (95% CI) | 1.000 (1.000–1.000) | 1.000 (1.000–1.000) | 0.875 (0.625–1.000) | 0.778 (0.500–1.000) | 1.000 (1.000–1.000) | 1.000 (1.000–1.000) |
| HR recall (95% CI) | 0.833 (0.768–0.900) | 0.809 (0.736–0.878) | 0.500 (0.167–0.800) | 0.444 (0.167–0.750) | 0.667 (0.000–1.000) | 0.667 (0.000–1.000) |

Ex3 excludes data that were missing three or more values, HR high risk, MR medium risk. Guangdong cohort had no missing values.

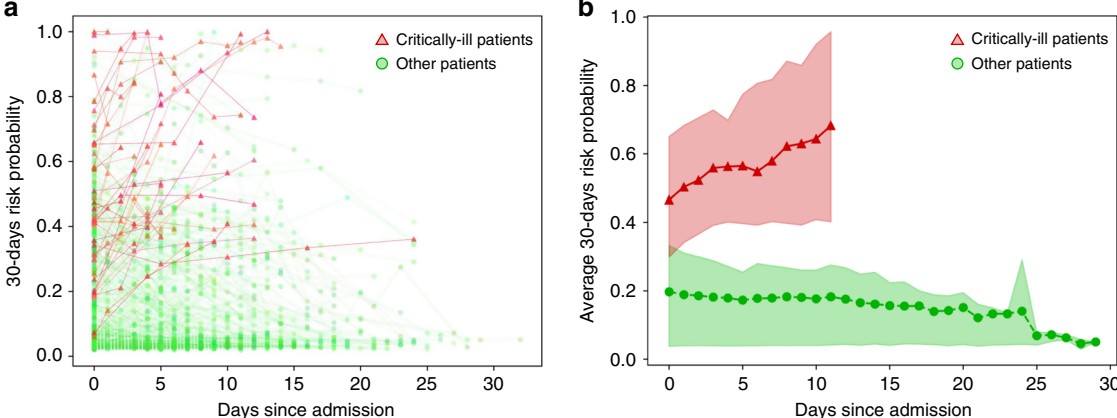

**Fig. 2 Trend of 30-days critically ill risk probability in the follow-up visit after admission.** Red lines with triangle markers are critically ill patients. Green lines with circle markers are other patients. **a** Visualization of trend of each individual. Each marker indicates a follow-up exam. For better visualization, line color has been slightly disturbed for each patient. **b** Average trend for different groups of patients. Colored area corresponds to the 25% and 75% of the risk probability.

Fig. 1c, d show the results of the entire external validation datasets and Ex3 datasets that excluded patients with more than three missing clinical features out of the ten required. The C-index of the entire dataset for the Wuhan, Hubei, and Guangdong cohorts were 0.878, 0.769, and 0.967, respectively. In the Ex3 dataset, the C-index for these cohorts were 0.890, 0.852, and 0.967, respectively.

**Risk monitoring**. Among the Wuhan cohort of 940 patients with dynamic data, 457 patients had follow-up exams (computed tomography (CT) and blood tests) after hospital admission. In addition to calculating the risk of developing critical illness at hospital admission, we also calculated the risk at follow-up exam times. As shown in Fig. 2, our model not only captures the risk of critical illness at admission but also can be used to monitor the trend of the risk during patients' hospital stay. The prediction performances of AUC and C-index at the follow-up exam time are 0.960 and 0.935, respectively, which are higher than those at the hospital admission (0.881 and 0.878, respectively). These results indicate that the clinical features better reflect the risk of critical illness as it draws closer to the event.

**Online patient triage tool**. Nomogram is a pictorial representation for depicting the association between clinical variables and the probabilities of clinical events such as critical illness, which provides an intuitive way to interpret the survival model[10]. We developed an online tool embedding a nomogram with our Deep Learning Survival Cox model at https://aihealthcare.tencent. com/COVID19-Triage_en.html. After a clinical staff fill in the online form with baseline clinical features, the tool returns a personalized nomogram, together with the probability of critical illness within 5, 10, and 30 days (Fig. 3).

## Discussion

All included variables were independently correlated with disease progression. Age is the most recognized risk factor for prognosis of COVID-19, with the most severe and fatal cases among patients over 60 years old. Respiratory tract symptoms, abnormalities in chest X-rays (compared with CT scans), and low lymphocyte ratios reflect the severity of the disease. Comorbidities, especially COPD and cancer, are strongly linked with the development of critical illness[11,12]. Similarly, age (over 60 years) and comorbid disease were also risk factors for poor outcome in severe acute respiratory syndrome (SARS) patients in 2003[13]. Compared with SARS-CoV and MERS-CoV, more deaths have been caused by multiple organ dysfunction syndrome rather than respiratory failure during COVID-19, which may be attributed to the widespread distribution of angiotensin-converting enzyme 2, the functional receptor for SARS-CoV-2, in multiple organs[14,15]. This explains why the blood test, such as lactate dehydrogenase, creatine kinase, will play a role in predicting critical illness.

CPH model is the traditional method for survival analysis and event prediction. However, it is a semiparametric model that assumes that a patient's risk of failure is a linear combination of the patient's clinical factors. The deep learning model is able to learn and infer high-order nonlinear associations between clinical covariates and patient outcomes in a fully data-driven manner. Furthermore, data augmentation strategies in deep learning can make the model more resilient to data noise and missing data, which commonly occurs in clinical datasets. The deep learning model can be also extended to incorporate time-dependent covariates such as vital signs and high-dimensional features such a CT or X-ray images.

Our model currently uses ten clinical variables, which are all common demographic and clinical characteristics, as well as laboratory results that are available at most hospitals. Despite this,

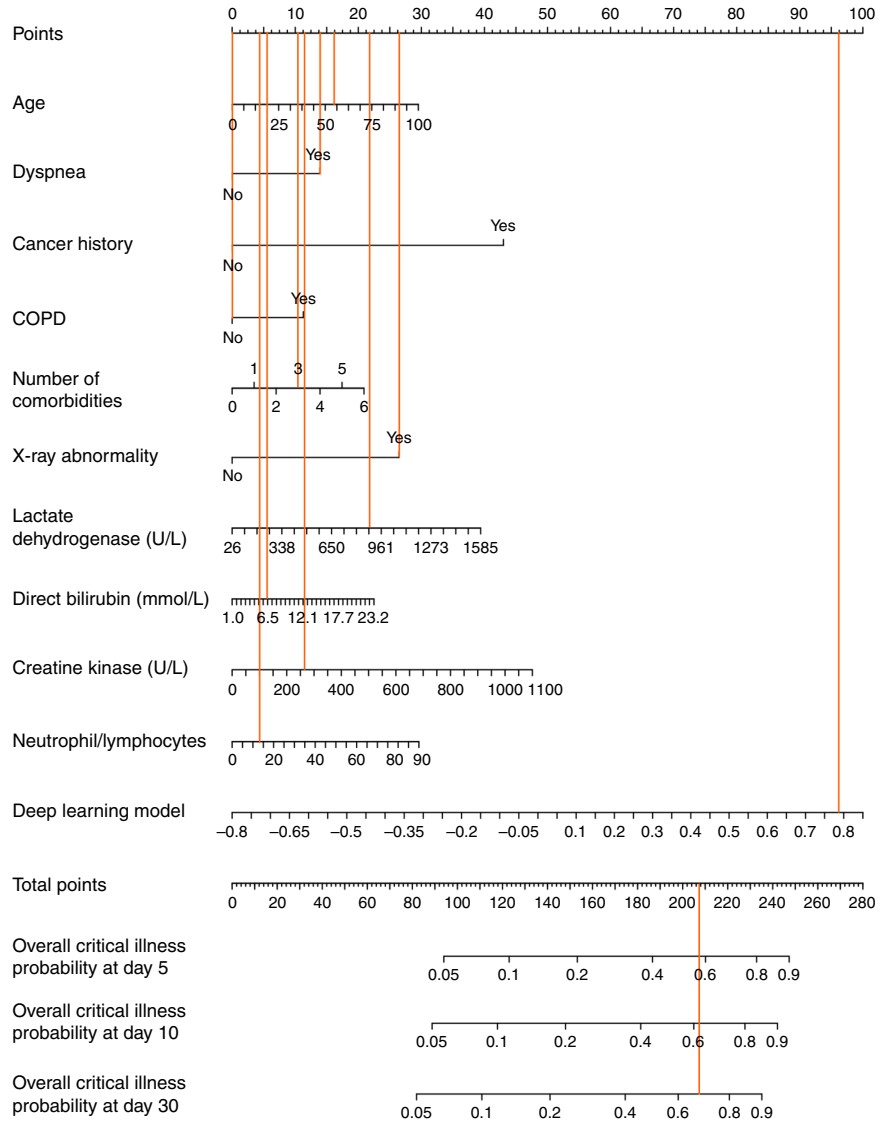

**Fig. 3 Nomogram of the Deep Learning Survival Cox model to triage COVID-19 patients.** The patient's total nomogram point is 209, overall critical illness probabilities are 0.58, 0.62, and 0.69 within 5, 10, and 30 days, respectively. The patient is triaged as high-risk.

more than 50% of our patients did not have all required values collected. Missing data can occur particularly with small or poorly equipped hospitals. Our model has a certain tolerance to missing data, as we still achieved high performance on the external validation set for cases missing 30% of the data. However, to take full advantage of this model, we recommend that all clinical features are collected at hospital admission. In real-world practice, missing data on some variables is inevitable. Therefore, missing data on less than three variables is allowed in our online calculation tool and the background can still provide a risk estimation based on deep learning imputation methods.

Our Deep Learning Survival Cox model demonstrated superior discriminating power compared with the classical Cox model, because it unravels the nonlinear relationships among complex clinical covariates and their hazards. To make clinically relevant comparison, we computed partial the area under the receiver operating characteristic curve (p-AUROC), where only the portion of the curve with sensitivity ≥0.8 was counted. The comparison between our deep learning survival Cox model and the classic Cox model is summarized in Supplementary Table 7. From the results, our proposed model is statistically better than

($p < 0.05$) the classic Cox model in terms of C-index and p-AUROC.

We investigated the false negatives in the external validation sets. Among the 106 critical cases, only 2 cases are classified as low risk. Both cases suffer from data missing and all the observed values land in the range of negative samples. For instance, both cases have no X-ray abnormality findings, no dyspnea, and no comorbidity including COPD and cancer history. Thus, these two cases are all outliers. Based on the observed values, we believe it is reasonable to classify them as low risk.

In our clinical experience, mild COVID-19 cases are generally self-limiting and it is the severe cases that require the most medical attention. Our proposed patient stratification tool has high clinical and economical value for COVID-19 disease management, particularly in light of the unusually rapid disease progression that can occur and the high mortality rate associated with critical illness. By submitting clinical information online, medical staff can triage patients at hospital admission using the predicted risk indicator and arrange patient treatment plans accordingly, ensuring patients receive treatment early and medical resources can be efficiently allocated. Based on the nature of

deep leaning, future prospective application and validation can help to further evolve this model.

## Methods

**Ethical approval**. This study was approved by the ethical review committee of the major included hospitals, who also waived the informed consent from patients.

**Data extraction and processing**. A team of experienced respiratory clinicians reviewed, abstracted, and cross-checked the data. Data were entered into a computerized database and cross-checked. Examination and treatment information was available and collected. The recent exposure history, clinical symptoms and signs, and laboratory findings upon admission were extracted from electronic medical records. Radiologic assessments, including chest X-ray or CT, were performed based on the documentation/description in medical charts or combined with, if imaging films were available, a review by our medical staff. Major disagreement between two reviewers was resolved by consultation with a third reviewer.

**Study design**. We performed a multivariate imputation by chained equation to fill in the missing data[16]. We employed a CPH with LASSO penalty to identify baseline clinical features that are associated with the later critical illness status. We then constructed a three-layer feedforward neural network using the selected features for survival modeling[7]. We designed a nomogram integrating the deep learning output as a patient triage tool at hospital admission[8]. According to the risk probability returned from the model, the patients are triaged into three groups: low, medium, and high risk of critical illness, at 95% sensitivity and 95% specificity, respectively. The C-index and AUC were evaluated on the validation cohort to assess the discriminative ability. We also compared this model with the CURB-6 model, which has been used in classification of community-acquired pneumonia cases[9]. All statistical tests were two sided and p-values < 0.05 indicated statistical significant.

**Data imputation**. We applied multivariate imputation via chained equations to impute the missing data[16]. The overall features were divided into three groups, numeric features, binary features (with two levels) and factor features (≥2 levels). For each kind of features, we applied different imputation methods. We used predictive mean matching to impute numeric features, logistic regression to impute binary variables and Bayesian polytomous regression to impute factor features. After data imputation, we normalized all features to 0 mean and 1 SD.

**Regularized Cox model with LASSO penalty**. We performed LASSO algorithm to select and sort the statistically significant clinical features[17]. We used critical illness as event in the analysis and the training cohort of 1590 patients and 74 clinical features for feature selection. We performed a tenfold cross-validation on the training set to calculate the weight of LASSO penalty (denoted as lambda). The lambda with 1 SE of the minimum partial likelihood deviance was used for feature selection.

**Feedforward neural network for survival modeling**. We constructed a three-layer feedforward neural network for survival modeling (namely deep survival model)[7]. The network architecture is illustrated in Supplementary Fig. 2. The ten selected features were fed into the network after data normalization. The network is composed by three fully connected layers including two hidden layers and one output layer. We empirically selected *tanh* as activation function. Output of the network is a single node, which predicts the risk score of developing critical illness event. If an event $i: E_i = 1$ happens before event $j: T_j > T_i$, then its risk score should be higher: $R_i > R_j$. Given this definition, the loss of the network is defined as following:

$$\text{Loss}(\theta) = - \sum_{i: E_i = 1} \left( h(x_i | \theta) - \log \sum_{j: T_j > T_i} e^{h(x_j | \theta)} \right) \quad (1)$$

where $\theta$ is the parameter of the model to be optimized and $h(x_i|\theta)$ is the risk score predicted by the network given input features $x_i$.

The network was optimized by gradient descending with gradients estimated by Adam optimizer. To avoid overfitting, dropout was applied after each layer during training. Hyperparameters including layer size, learning rate, dropout rate, and training epochs were optimized by Bayesian Hyperparameters Optimization[18]. The final optimized parameters are listed in Supplementary Table 6. The final model was obtained by training the network with the optimal hyperparameters on the whole training set.

**Deep learning survival Cox model**. We combined the ten features selected by the LASSO Cox model with the output of our deep survival model and constructed an integrated Cox model (named Deep Learning Survival Cox model). We performed ridge regression with Cox loss on the same training set described above with a tenfold cross-validation.

**Reporting summary**. Further information on research design is available in the Nature Research Reporting Summary linked to this article.

## Data availability

The datasets generated during and/or analyzed during the current study are not publicly available due to the confidential policy of National Health Commission of China, but are available from the corresponding author Jianxing He upon reasonable request. In addition, this database is open for validation of results of other future studies worldwide, through collaboration with the staff of the China Clinical Research Center for Respiratory Disease.

## Code availability

The code being used in the current study for developing the algorithm is provided at https://github.com/cojocchen/covid19_critically_ill.

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

## Acknowledgements

This study is supported by China National Science Foundation (Grant numbers 81871893 and 81903421), Key Project of Guangzhou Scientific Research Project (Grant number 201804020030), High-level university construction project of Guangzhou medical university (Grant numbers 20182737, 201721007, 201715907, and 2017160107), National key R & D Program (Grant numbers 2017YFC0907903 and 2017YFC0112704), and the Guangdong high-level hospital construction "reaching peak" plan. This work was partially supported by the Key Area Research and Development Program of Guangdong

Province, China (number 2018B010111001), National Key Research and Development Project (2018YFC2000702), and Science and Technology Program of Shenzhen, China (number ZDSYS201802021814180). We thank the hospital staff (see Supplementary Appendix for a full list of the staff) for their efforts in collecting the information. We are indebted to the coordination of Drs. Zong-jiu Zhang, Ya-hui Jiao, Bin Du, Xin-qiang Gao and Tao Wei (National Health Commission), Yu-fei Duan and Zhi-ling Zhao (Health Commission of Guangdong Province), Yi-min Li, Zi-jing Liang, Nuo-fu Zhang, Shi-yue Li, Qing-hui Huang, Wen-xi Huang, and Ming Li (Guangzhou Institute of Respiratory Health), which greatly facilitate the collection of patient's data. Special thanks are given to the statistical team members Professor Zheng Chen, Drs. Dong Han, Li Li, Zheng Chen, Zhi-ying Zhan, Jin-jian Chen, Li-jun Xu, and Xiao-han Xu (State Key Laboratory of Organ Failure Research, Department of Biostatistics, Guangdong Provincial Key Laboratory of Tropical Disease Research, School of Public Health, Southern Medical University). We also thank Li-qiang Wang, Wei-peng Cai, Zi-sheng Chen (the sixth affiliated hospital of Guangzhou Medical University), Chang-xing Ou, Xiao-min Peng, Si-ni Cui, Yuan Wang, Mou Zeng, Xin Hao, Qi-hua He, Jing-pei Li, Xu-kai Li, Wei Wang, Li-min Ou, Ya-lei Zhang, Jing-wei Liu, Xin-guo Xiong, Wei-juna Shi, San-mei Yu, Run-dong Qin, Si-yang Yao, Bo-meng Zhang, Xiao-hong Xie, Zhan-hong Xie, Wan-di Wang, Xiao-xian Zhang, Hui-yin Xu, Zi-qing Zhou, Ying Jiang, Ni Liu, Jing-jing Yuan, Zheng Zhu, Jie-xia Zhang, Hong-hao Li, Wei-hua Huang, Lu-lin Wang, Jie-ying Li, Li-fen Gao, Jia-bo Gao, Cai-chen Li, Xue-wei Chen, Jia-bo Gao, Ming-shan Xue, Shou-xie Huang, Jia-man Tang, Wei-li Gu, and Jin-lin Wang (Guangzhou Institute of Respiratory Health) for their dedication to data entry and verification. Finally, we thank all the patients who donate their data for analysis and the medical staffs working in the front line.

## Author contributions

W.L.: write manuscript, design study, data collection, processing, and analysis. J. Yao: write manuscript, design experiment, and data analysis. A.C.: design study, data processing, and collection. Q.L.: data collection and processing. S.W.: write manuscript, design study. M.Z.: write manuscript, design study. Y.L.: administrative support, write manuscript. J. Liu: administrative support, data collection, and processing. J. Lu: data collection and processing. H.L.: data processing and analysis. G.C.: data collection and processing. H.G.: data collection and processing. J.G.: data collection and processing. R.Z.: design study, data collection, and processing. L.O.: write manuscript, data processing, and analysis. N. Zhou: implement model, design experiment, data analysis, and write manuscript. H.C.: implement model, design experiment, data analysis, and write manuscript. F.Y.: implement model, design experiment, data analysis, and write manuscript. X.H.: design experiment and write manuscript. W.T.: design experiment. W.H.: administrative support, design experiment, and write manuscript. W.G.: write manuscript, data processing, and analysis. Z.C.: data processing. Y.Z.: data processing. S. Ling: data collection and processing. Y.X.: data collection and processing. W.W.: data collection and processing. S. Li: administrative support, design study, data collection, and processing. L.L.: data collection and processing. N. Zhang: administrative support. N. Zhong: administrative support, design study, data collection, and write manuscript. J. Huang: modeling problem, design experiment, and write manuscript. J. He: administrative support, design study, data collection, and write manuscript.

## Competing interests

The authors declare no competing interests.
