## [Peer Review File · Nature Communications]

Reviewers' Comments:

Reviewer #1:

None

Reviewer #2:

Remarks to the Author:

I think the Authors have made a comprehensive attempt to answer all the Reviewers' Comments as best they can - the proof of this will be when it can be used in real-life situations - which, like any other application in clinical medicine, whether drugs treatments, artificial limbs, diagnostic procedures, development and application of guidelines, etc. will work better in some patients in some hospitals in some populations than others - but this is not a reason to just stop trying.

Reviewer #3:

None

Reviewers' Comments:

Reviewer #2:

Remarks to the Author:

I think the Authors have made a comprehensive attempt to answer all the Reviewers' Comments as best they can - the proof of this will be when it can be used in real-life situations - which, like any other application in clinical medicine, whether drugs treatments, artificial limbs, diagnostic procedures, development and application of guidelines, etc. will work better in some patients in some hospitals in some populations than others - but this is not a reason to just stop trying.

R: thank you for the positive comments. We have developed an online tool that can be put in used in real-life situation. We also open source our tool that anyone can improve it.